# Growth faltering or deceleration toward target height: Linear growth interpretation using WHO growth standard 2006 for Indonesian children

**Annang Giri Moelyo[1]\*, Mei Neni Sitaresmi[2], Madarina Julia[2]**

**1** Department of Child Health, Faculty of Medicine, Universitas Sebelas Maret/Dr. Moewardi Hospital, Surakarta, Indonesia, **2** Department of Child Health, Faculty of Medicine, Public Health and Nursing, Universitas Gadjah Mada/Dr. Sardjito Hospital, Yogyakarta, Indonesia

☯ These authors contributed equally to this work.
\* annanggm73@gmail.com

## Abstract

### Objective

When referred to the WHO Growth Standards 2006, children in many developing countries showed growth faltering in childhood. A previous study showed that the faltering affected the whole population, not only the disadvantaged ones. We aimed to look for an alternative explanation for this universal decline in length or height-for-age z-scores (HAZ), as lengths/ heights of Indonesian children were compared to the WHO growth standard 2006: Is it a faltering of growth or is it a deceleration toward target height.

### Methods

We used data on age, gender, height, BMI, parental height and education, household socioeconomic status, and place of residence of children < 5 years old collected by the Indonesia Family Life Survey (IFLS) in 1993, 2000, 2007, and 2014. HAZ was calculated according to the WHO 2006 growth standard. Target heights were calculated from parental heights and converted to target height z-scores (THz). Discrepancies between the two values were used to show the children's growth patterns in relation to their target heights across ages.

### Results

The study included 11,241 parent-child pairs from four surveys. At birth, infants were around 1.50 z-scores longer than their THz. However, at two years of age, the discrepancies were almost zero. At 2–5 years old, the discrepancies stayed at the same level. The patterns were similar regardless of the position of the target heights among the height distribution, at the upper or the lower part.

**Data availability statement:** The datasets generated and/or analyzed during the current study are available at https://doi.org/10.6084/m9.figshare.23726754

**Funding:** The author(s) received no specific funding for this work.

**Competing interests:** The authors have declared that no competing interests exist.

## Conclusion

We observed a deceleration toward target height, not growth faltering, in the first two years of life of Indonesian children when the WHO Growth Standard 2006 was used as the reference.

## Introduction

The introduction to the WHO Growth Standards 2006 left many developing countries challenged by issues of growth faltering and stunting early in childhood. From an epidemiological, biological, and parental standpoint, recent studies have questioned the WHO Growth Standards' one-size-fits-all approach. Arguments arise from issues with normal variability and disparities among countries (social determinants of health, environmental factors, and genetic composition) [1,2].

When converted to WHO growth standards 2006 standard deviation or z-scores, the lengths/ heights of young children in low- and middle-income countries decreased dramatically until the age of 24 months [3]. Since then, many studies, mostly focused on environmental-level risk factors, i.e., individuals (maternal, fetal, infancy, or childhood) or household level, were performed [4–6].

In 2017, Roth et al. suggested that, instead of focusing on individual or household-level risk factors, to look for community-level determinants of childhood HAZ trajectories. His suggestion was based on his observation that the declines in mean HAZ were due to a downward shift in the entire HAZ distribution, i.e., children across the HAZ spectrum grew slower than the international standard. This finding may mean that the whole population experienced growth faltering, and a child with a relatively low HAZ was not more likely to have faltered than their taller same-age peers [3].

Indonesia is one of the countries bothered by this growth faltering/ stunting issue. Although the prevalence of 'stunting' has improved in recent years; still, around a fourth or almost a third of Indonesian under five years old children would be diagnosed as stunting when the WHO Growth Standards 2006 were used as the reference [7,8]. In fact, many studies have criticized the suitability of the WHO Growth Standards 2006 to define growth faltering or stunting in many countries, including Indonesia [2,9–11].

When a child's growth decelerates against the growth chart, people will interpret it as growth faltering. People tend to forget the other possible explanation for this downward deflection, i.e., the classic concept of canalization or the process of deflection toward target height. Target height, defined as the predicted final height of a child based on the parents' heights, may cause acceleration or deceleration of growth according to where the length of the child at a certain age compared to the heights of his/ her parents. A child born shorter than his/ her target height is expected to accelerate or catch up, while on the other hand, a child born taller is expected to decelerate or catch down [12–15]. To see whether a child's length z-score is approaching his/ her target height, we should calculate his/ her target height z-score. Target height z-scores can be calculated using z-scores of the sex-corrected mid-parental height (MPH) or the mean of the father's and mother's height-for-age z-scores [12].

In this paper, using multiple cross-sectional surveys of Indonesian parent-child pairs, we hypothesized that the older the child, the closer he or she will be to his or her target height. Further, we also hypothesized that the 'universal' growth deceleration observed when the growth of Indonesian children was compared to the WHO growth standard 2006 was not growth faltering but a community or a population-level pattern of growth trajectory toward target height.

## Methods

### Participants and data sources

Data were obtained from the Indonesia Family Life Survey (IFLS) surveys that are freely available upon request at www.rand.org. The surveys were conducted by the Research and Development (RAND) Corporation United States in collaboration with Universitas Indonesia and Universitas Gadjah Mada. It was done in five waves: 1993 (wave 1), 1997–1998 (wave 2), 2000 (wave 3), 2007 (wave 4), and 2014 (wave 5) [16–19].

The IFLS data are freely available on https://www.rand.org/about.html with the help of a survey from Lembaga Demografi, Faculty of Economics and Business, University of Indonesia (LD FEB UI, http://ldfebui.org/en/profil-ld/) for IFLS 1 and 2; and a private Indonesian research agency, Surveymeter (http://surveymeter.org/page/26/tentang-kami), for IFLS 3, 4, and 5. The IFLS surveys and their procedures have been reviewed and approved by IRBs (Institutional Review Boards) in the United States (at RAND) and in Indonesia at the University of Gadjah Mada for IFLS3, IFLS4, and IFLS5 and at the University of Indonesia for IFLS1 and IFLS2 (https://www.rand.org/well-being/social-and-behavioral-policy/data/FLS/IFLS.html). The ethical clearance number is s0064-06–01-CR01 from RAND's Human Subjects Protection Committee (RAND's IRB) for IFLS5 (https://www.rand.org/well-being/social-and-behavioral-policy/data/FLS/IFLS/datanotes.html#ethical). All methods were carried out in accordance with relevant guidelines and regulations. All participants in the IFLS survey gave written informed consent that was obtained prior to data collection (https://www.rand.org/content/dam/rand/pubs/working_papers/WR1100/WR1143z3/RAND_WR1143z3.pdf) [16–19].

The IFLS was done in 13 of the country's 26 provinces and represented around 83% of the Indonesian population. It used multistage sampling to sample households. The survey randomly selected 321 areas in the 13 provinces with intentional oversampling of smaller provinces. In the selected areas, 20 or 30 households were randomly selected from each rural or urban enumeration area, respectively. The sample comprised of more than 7,200 households and 22,000 individuals [19]. All participants, their offspring, and split-off households from wave 1 were re-contacted to join waves 2, 3, 4 and 5; a more than 90% re-contact rate was observed in each subsequent wave [16–19].

We downloaded the data in October 2019. This study included data from subjects less than five years of age. Data on age (months), body length or height (cm, recumbent length in children younger than 24 months and standing height in older children), body weight (kg), gender, parental height (cm, father and mother's height), parental education, household socio-economic status, and residences (living in the rural or urban area, living in the island of Java - Bali or in other islands) were obtained from waves 1 (1993–1994), 3 (2000), 4 (2007) and 5 (2014). We excluded data from wave two because it was too close to wave 3 to get a meaningful trend observation [16–19].

In every wave, anthropometric measurement and data collection were performed using a similar rigid protocol and questionnaires. Every data collection team comprised two nurses who underwent didactic and hands-on anthropometric training before data collection. Their birth certificates verified information on the children's date of birth and gender. Information on the number of household members, food and non-food expenditures, and the level of education of parents were provided by the household head. Age was calculated from the date of birth and the date of the anthropometric measurement (in months) [16–19].

Children younger than two years were measured lying down using SECA plastic length board to the nearest millimetre. Children 2 years old or above and the adults were measured standing up using Short Measuring Boards Model 420. Weights were measured using SECA Model 770 scales (SECA, Los Angeles, CA, USA) [7,20].

A computer-assisted program was used to check the completeness of data and the possibility of errors. Constant supervision was performed to ensure that the data collection protocol was closely followed [16–19].

Lengths or heights and BMI (body mass indexes) were converted to length or height z-scores (HAZ) and BMI-for-age z-score (BAZ) based on the WHO Child Growth Standard 2006 [21]. We used the zanthro command in Stata version 14.0 to calculate the standard deviation z-scores [22]. We excluded outliers, i.e., less than -4 and over +4 standard deviation z-scores. [22] BAZ (BMI-for-age z-score) was used to define nutritional status into underweight (<-2 SD), normal weight (-2 to less than +2 SD), and overweight–obese (+2 SD or above).

Target height (TH) was calculated using the following formula: father's height + mother's height + d)/ 2 cm (boys); father's height + mother's height – d)/ 2 cm (girls). The "d" was calculated from the difference between the mean of the father's and the mother's heights in each wave. Target height z-score (THz) was calculated using WHO Growth Reference 2007 for school-aged children and adolescents by the child's gender for age z-scores at 19 years, assuming that parental heights remained the same after the age of 19 years. We used WHO Growth Reference 2007 as the reference for target heights because despite being based on US-NHANES children, the WHO Growth Reference 2007 was already designed to smoothly conform to WHO standard 2006, to be regarded as the continuation of WHO standard 2006 [23].

We used the zanthro command in Stata version 14.0 to calculate the standard deviation z-score at 19 years. The discrepancy between the current HAZ of the children and the target height z-score (THz) was calculated by HAZ minus THz [24]. The positive results showed that HAZ was higher than THz and vice versa.

Several characteristics defined by the Indonesian Bureau of Statistics have been used to distinguish between rural and urban areas (i.e., the size of the area, the population's size and density, the main occupation of its people, and the availability of public facilities).Household food expenditure share is the monthly food expenditure divided by the total monthly expenditure in a household. The higher the share, the poorer the subjects' households [25]. Basic education in parental education was defined as less than nine years of formal education.

## Statistical analysis

We used scatter plots and LOWESS smoothing line (locally weighted scatterplot smoothing) commands in Stata 14.0 to show discrepancies between the child's current HAZ with THz in each wave, among target height z-score subgroups, and between urban and rural areas. Differences between groups were analyzed using one-way ANOVA or chi-squares, with significant levels of < 0.05. All analyses were done with Stata version 14.0.

## Results

The study included data from 11,241 children in four surveys (waves 1, 3, 4, and 5). The recent waves showed improvements in the children's HAZ, household expenditure shares, and parental education (Table 1). We also observed improvement in TH and THz (Table 2).

Fig 1 shows the discrepancy between the children's HAZ and their THz among waves. A decline in the discrepancy between the children's HAZ and their THz was observed from birth to the age of approximately two years. At birth, infants were around 1.50 z-scores longer than their THz. However, at two years of age, the discrepancy was almost zero. Similar patterns were observed across waves. Supplementary Table 1 showed in detail the diminishing discrepancies across ages and waves. At the age of 21–24 months on wards the discrepancies were around zero.

**Table 1. Characteristics of the subjects.**

| | Wave 1 (1993) | Wave 2 (2000) | Wave 3 (2007) | Wave 4 (2014) |
|---|---|---|---|---|
| | N = 1,636 | N = 2,741 | N = 3,344 | N = 3,520 |
| Age groups (n,%) | | | | |
| • 0–<2 | 610 (37.3%) | 1,182 (43.1%) | 1,288 (38.5%) | 1,371 (38.9%) |
| • 2–<5 | 1,026 (62.7%) | 1,559 (56.9%) | 2,056 (61.5%) | 2,149 (61.1%) |
| Gender (n, %) | | | | |
| • Girls | 765 (46.8%) | 1,338 (48.8%) | 1,616 (48.3%) | 1,702 (48.4%) |
| • Boys | 871 (53.2%) | 1,403 (51.2%) | 1,728 (51.7%) | 1,818 (51.6%) |
| HAZ (mean, SD) | -1.78 (1.44) | -1.55 (1.46) | -1.42 (1.47) | -1.42 (1.38) |
| BAZ (mean, SD) | -0.41 (1.26) | -0.24 (1.37) | -0.09 (1.44) | -0.17 (1.40) |
| Nutritional status (BAZ) (n,%) | | | | |
| • underweight (<-2SD) | 141 (8.6%) | 244 (8.9%) | 274 (8.2%) | 263 (7.5%) |
| • normal weight (-2SD to less than +2 SD) | 1,443 (88.2%) | 2,345 (85.6%) | 2,831 (84.7%) | 3,009 (85.5%) |
| • overweight–obese (±2SD or above) | 52 (3.2%) | 152 (5.5%) | 239 (7.1%) | 248 (7.0%) |
| Number of household member | 5 (4-7) | 6 (4-8) | 5 (4-7) | 5 (4-8) |
| Household expenditure share (mean, SD) | 0.33 (0.23) | 0.25 (0.19) | 0.21 (0.16) | 0.19 (0.15) |
| Father's education (n,%) | | | | |
| basic education | 1,606 (98.2%) | 1,822 (66.5%) | 1,988 (59.4%) | 1,924 (54.7%) |
| high education | 30 (1.8%) | 919 (33.5%) | 1,356 (40.6%) | 1,596 (45.3%) |
| Mother's education (n,%) | | | | |
| basic education | 1,630 (99.6%) | 2,023 (73.8%) | 2,158 (64.5%) | 2,018 (57.3%) |
| high education | 6 (0.4%) | 718 (26.2%) | 1,186 (35.5%) | 1,502 (42.7%) |
| Islands (n,%) | | | | |
| Outer Java-Bali islands | 669 (40.9%) | 1,036 (37.8%) | 1,409 (42.1%) | 1,714 (48.7%) |
| Java-Bali islands | 967 (59.1%) | 1,705 (62.2%) | 1,935 (57.9%) | 1,806 (51.3%) |
| Residency (n,%) | | | | |
| rural | 837 (51.2%) | 1,496 (54.6%) | 1,628 (48.7%) | 1,587 (45.1%) |
| urban | 799 (48.8%) | 1,245 (45.4%) | 1,716 (51.3%) | 1,933 (54.9%) |

*HAZ: height-for-age z-score; BAZ: body mass index-for-age z-score; *one-way ANOVA test or chi-square test*

**Table 2. Parental heights and Target Heights (TH).**

| | Wave 1 (1993) | Wave 2 (2000) | Wave 3 (2007) | Wave 4 (2014) |
|---|---|---|---|---|
| | N = 1,636 | N = 2,741 | N = 3,344 | N = 3,520 |
| father's height (cm, SD) | 161.50 (5.75) | 161.70 (6.07) | 162.13 (5.92) | 162.71 (6.16) |
| father's HAZ (SD) | -2.06 (0.79) | -2.03 (0.83) | -1.98 (0.81) | -1.90 (0.84) |
| mother's height (cm, SD) | 150.21 (5.18) | 150.56 (5.36) | 151.24 (5.29) | 151.20 (5.39) |
| mother's HAZ (SD) | -1.98 (0.79) | -1.92 (0.82) | -1.82 (0.81) | -1.83 (0.82) |
| difference between father and mother's height (cm, SD) | 11.29 (7.25) | 11.14 (7.18) | 10.88 (7.29) | 11.51(7.39) |
| TH in boys (cm, SD) | 161.31 (4.11) | 161.74 (4.55) | 162.20 (4.27) | 162.51 (4.46) |
| TH z-score in boys (SD) | -2.09 (0.56) | -2.03 (0.62) | -1.96 (0.58) | -1.92 (0.61) |
| TH in girls (cm, SD) | 150.40 (4.07) | 150.53 (4.36) | 151.17 (4.29) | 151.40 (4.45) |
| TH z-score in girls (SD) | -1.95 (0.62) | -1.93 (0.67) | -1.83 (0.66) | -1.80 (0.68) |

*TH (target height) = (father's height + mother's height +/- 11)/2 cm (+: boys; -:girls)*

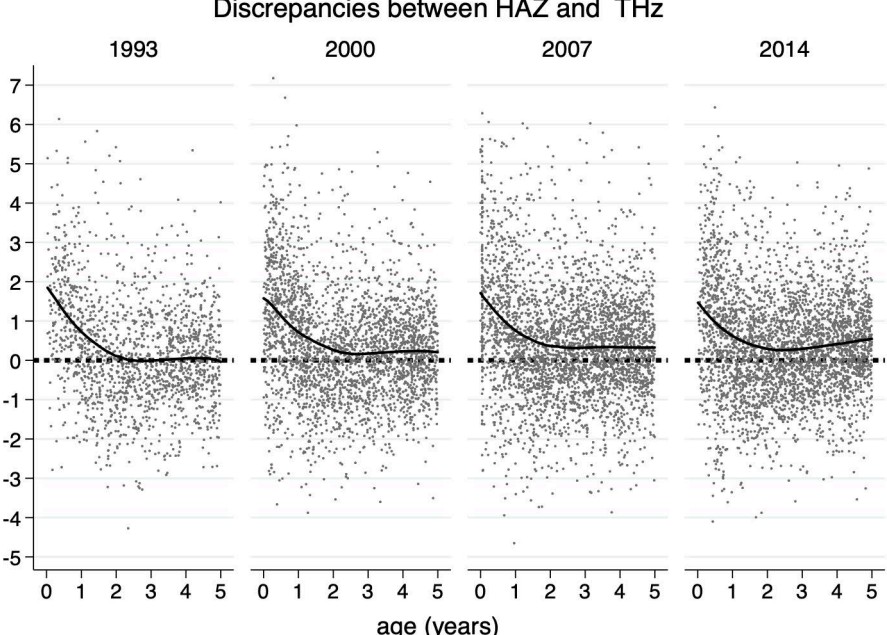

**Fig 1. Discrepancies between the children's height-for-age-z-scores (HAZ) and their target height-z-scores (THz) by waves.**

In Fig 2, we divided THz into four subgroups, i.e. THz less than -3, $-3 \leq$ THz < -2; -2 $\leq$ THz < -1, and THz $\geq$ -1). The figure showed that the growth of children whose parents were at the upper part of the height distribution, i.e., children whose THz were -1 or above, exhibited a similar phenomenon to children whose parents were at the lower part of the height distribution, whose THz were less than -3. All members of the population grew toward their target height. However, interestingly, the lower the THz, the bigger was the positive discrepancy between HAZ and THz, meaning children of relatively shorter parents had the potential to be taller than their parents, but not children of relatively taller parents.

Supplementary Fig 1 (S1 Fig) compares the decline in the discrepancy between rural and urban children. Rural children showed a sharper decline; they were even shorter than their target height at the age of 2–3 years in wave 1 (1993–1994). Children living in the rural areas consistently had a lower discrepancy between HAZ and THz, although the gap between the urban and the rural areas narrowed in the recent surveys. The z-scores gap between the urban and the rural areas in 2–5 years old children were 0.43 (95% CI: 0.28–0.58); 0.33 (95% CI: 0.21–0.45); 0.25 (95% CI: 0.14–0.36); and 0.18 (95% CI: 0.08–0.28) in 1993, 2000, 2007, and 2014, respectively.

Supplementary Table 2 (S2 Table) shows the multiple linear regression of THz and other contributing factors to HAZ among three age groups of subjects: 0- < 6 months, 6- < 24 months, and $\geq$ 2 years. The correlation coefficient between THz and HAZ increased from 0.37 to 0.52, and to 0.59 as the child grew older, meaning the older the child, the larger was the contribution of THz to his or her current HAZ ($p < 0.05$). At lower magnitude, the contribution of environmental factors to HAZ also increased. Total adjusted $R^2$ increased from 0.13 at the age below 6 months to 0.17 in children aged 2–5 years.

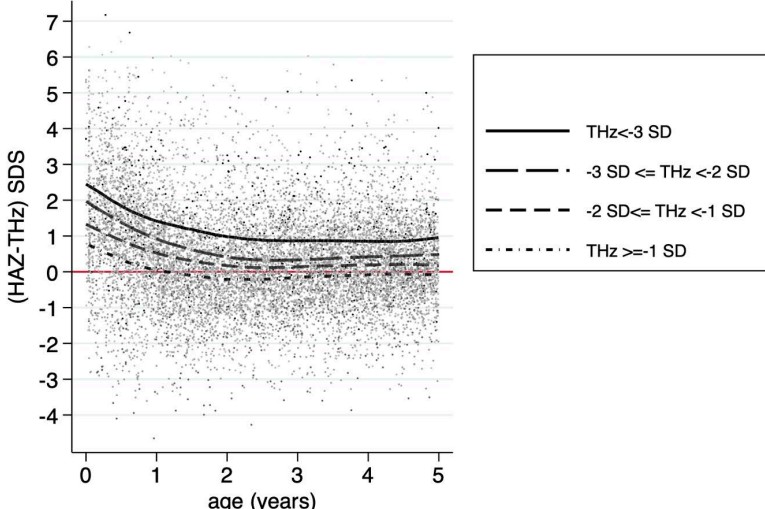

**Fig 2. The discrepancy between HAZ (height-for-age z-score) and THz (target height z-score) SDS among target height z-score subgroups.**

## Discussion

Our study observed that, compared to the WHO Growth Standard 2006, the HAZ of Indonesian infants declined, approaching their target height z-score (THz). At around two years of age, the discrepancies between HAZ and THz were at their lowest. This finding is important because distinguishing between faltering of growth related to environmental offenses, either due to lack of nutrition or illnesses, with a natural adaptation to target height, will bring considerable consequences to clinical approach and public health policy. Of course, not many things can be done for the latter phenomenon.

Fig 1 shows that children are typically born at a z-score greater than predicted by their parents height, and they then approach the predicted z-score as they get older. However, the more recent waves showed that the height of the children progressively exceeded the predicted heights. These data suggested that, besides the secular trend increase in height as had been reported before [25], both the intrauterine growth and the post-natal condition have improved since the time the parents were children, as would be expected.

Our previous study using the same population showed that, albeit small, adults in Indonesia underwent a secular increase in height [25]. As a child's target height is calculated from parental heights, the secular increase in adult heights will be naturally followed by an increase in target heights (Table 2). This, further, as our study observed, is followed by the increase in HAZ of the children, as seen in Table 1. This finding supported a previous study, which showed that the secular height trends of children corresponded to those in adults. Further, in line with what we observed, it had also observed that the secular trend toward adult height had been evident by as early as one-and-a-half to two years of age [26].

Our study also observed a fascinating phenomenon: while the growth of children whose parents were at the tallest part of the distribution was similar to those at the lower part of the distribution, relatively shorter parents had children whose HAZ were taller than their THz, but not children of the relatively taller parents. It seemed that, although THz still guided the course of how a child would grow, an inclination toward better growth, or at least regression toward the mean, was observed in children whose parents were shorter [27].

Failure to consider parental height when diagnosing growth faltering may lead to over-estimating the prevalence of growth failure [1,28,29]. Worse still, the consequence can be enormous when this natural phenomenon is considered as a nutritional deficiency. These infants were at risk of being provided with excess food, leading to later obesity and metabolic problems. Implications for public health policy may include opposition to the exclusive breastfeeding policy as well as providing supplementary nutritional program for infants who are not in need. Apart from spending too much resources pointlessly, such policies would potentially lead to increased prevalence of childhood and adulthood obesity with their metabolic consequences. It is importance to take a holistic approach when interpreting children whose decrease growth. It needs careful and comprehensive assessments that consider various contributing factors, such as nutritional Intake, infection/ illness, socio-economic and environment factors.

A previous study by Roth et al. showed that the decline in HAZ was associated with a downward shift in the overall HAZ distribution, which was not only the decrease of the means but also in the fifth and 95th percentiles. He suggested that this universal decline must be related to a factor or factors that affected the whole population, not only the disadvantaged individuals or households in that population [3]. We propose that 'the factor', at least for Indonesia, is the target height. We have also shown that the contribution of target heights to a child's current height was larger than that of other individuals or household predictors. Our findings showed that better socioeconomic status showed better heights; however, the deceleration patterns of height z-scores were similar.

In a previous study discussing the timing of the transition from infant growth to childhood growth pattern, the transition was characterized by a switch from a predominantly decelerating growth of infancy into a steadier state of childhood growth. Different growth velocity illustrated the timing of the transition [30]. It was noticed that the timing of the transition was approximately at the age of 24 months [3,30].

## Limitations of the study

Of course, the best way to show whether the children will grow toward their parents' height is to follow their growth until they reach maturity. However, such information will be challenging to obtain in a developing country like Indonesia, where continuous population-level data gathering is not in place. Using cross-sectional information on the heights of thousands of Indonesian parent-child (of various ages and various periods) pairs, we expect to show the children's height trajectories in relation to their parents' height.

This study was not intended to show individual predictors of heights. Our regression analysis showed that only 13–17% of the variability in HAZ could be explained by the models made from the available data. Regression to the mean [27], and SEPE (social economy political emotional) factors [31], and many other predictors of height, most of them are still yet to be identified, should be our research priority in order to better explain variability in heights.

## Conclusion

We observed a deceleration toward target height in the first two years of life of Indonesian children when the WHO Growth Standard 2006 was used as the reference. The deceleration was observed in the whole population, regardless of whether the target height was at the top or the lower part of the height distribution. The entire population growth trajectory declined in unison. These findings contribute to the broader discourse on growth assessment with the WHO Growth Standard 2006. This finding is important because distinguishing between growth faltering from normal adaptation toward target height will have substantial implications for public health policy.

## Supporting information

**S1 Fig. Discrepancies between the children's height-for-age-z-scores (HAZ) and their target height-z-scores (THz) between rural and urban areas by waves.**
(TIF)

**S1 Table. Discrepancies between the children's length or height-for-age z-scores (HAZ) and their target height z-scores (THz) in several age groups based on waves.**
(DOCX)

**S2 Table. (2a, 2b and 2c).** Univariate and multivariate analysis on predictors of height-for-age z-score (HAZ) in children aged 0-<6 months, 6-<24 months, and >24 months.
(DOCX)

## Author contributions

**Conceptualization:** Annang Giri Moelyo, Mei Neni Sitaresmi, Madarina Julia.

**Data curation:** Annang Giri Moelyo, Mei Neni Sitaresmi, Madarina Julia.

**Formal analysis:** Annang Giri Moelyo, Mei Neni Sitaresmi, Madarina Julia.

**Investigation:** Annang Giri Moelyo.

**Methodology:** Annang Giri Moelyo, Mei Neni Sitaresmi, Madarina Julia.

**Resources:** Annang Giri Moelyo.

**Supervision:** Annang Giri Moelyo, Mei Neni Sitaresmi, Madarina Julia.

**Validation:** Annang Giri Moelyo, Mei Neni Sitaresmi, Madarina Julia.

**Visualization:** Annang Giri Moelyo, Madarina Julia.

**Writing – original draft:** Annang Giri Moelyo.

**Writing – review & editing:** Annang Giri Moelyo, Mei Neni Sitaresmi, Madarina Julia.

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
