## [Decision Letter · Decision Letter 0]

12 Sep 2023

PONE-D-23-23734Growth faltering or deceleration toward genetic target height: 

linear growth interpretation using WHO growth standard 2006 

for Indonesian childrenPLOS ONE

Dear Dr. Moelyo,

Thank you for submitting your manuscript to PLOS ONE. After careful consideration, we feel that it has merit but does not fully meet PLOS ONE’s publication criteria as it currently stands. Therefore, we invite you to submit a revised version of the manuscript that addresses the points raised during the review process.

We look forward to receiving your revised manuscript.

Kind regards,

Rebecca Evelyn Jones-Antwi, MPH

Academic Editor

PLOS ONE

Journal Requirements:

Additional Editor Comments:

We thank the authors for the manuscript for potential publication. We request a major revision. Below you will find some very thoughtful and detailed comments by two reviewers that we request to help improve the paper.

Reviewers' comments:

Reviewer's Responses to Questions

**Comments to the Author**

1. Is the manuscript technically sound, and do the data support the conclusions?

Reviewer #1: No

Reviewer #2: Yes

2. Has the statistical analysis been performed appropriately and rigorously? 

Reviewer #1: No

Reviewer #2: Yes

3. Have the authors made all data underlying the findings in their manuscript fully available?

Reviewer #1: Yes

Reviewer #2: Yes

4. Is the manuscript presented in an intelligible fashion and written in standard English?

Reviewer #1: Yes

Reviewer #2: Yes

5. Review Comments to the Author

Reviewer #1: This analysis starts with the premise that in Indonesia as the case of other low- and middle-income countries, there is a downward shift of the HAZ distribution across all population of children under 5 years of age when compared to the 2006 WHO growth standards. The authors hypothesized that this negative shift of the distribution is not a reflection of growth faltering, but a deceleration towards their genetic target height (their parents attained height in adult life). It is scientifically sound to explore the causes and/or possible factors behind this population shift of the HAZ distribution among children under 5 in Indonesia, however, the evidence presented is not strong and clear enough to support that deceleration towards the genetic target height is the reason behind this distribution shift, and to rule out growth faltering across the whole population as the cause, as has been suggested in previous analysis (Reference 3: Roth DE, Krishna A, Leung M, Shi J, Bassani DG, Barros AJD. Early childhood linear growth faltering in low-income and middle-income countries as a whole-population condition: analysis of 179 Demographic and Health Surveys from 64 countries (1993-2015). Lancet Glob Health. 2017 Dec;5(12):e1249-e1257.).

Below, I raise some specific comments and questions to clarify the rationale behind the hypothesis, the methods applied to test this hypothesis, and interpretation of findings.

Major comments

Introduction

• Based on the text from the introduction and findings from Roth et al, 2017 study that authors quote at stating that community-level factors might be behind the downward shifts of HAZ distribution in the entire population. It is not clear for me, why authors hypothesized that genetic target height, an individual-level factor, might be the factor behind the shifts of the HAZ distribution among Indonesian children.

• Page 3: “The classic concept of growth acceleration or deceleration toward their genetic target height in the first few years of life might become a plausible explanation for this 'universal' growth faltering.” Could authors talk a little bit more on the genetic target height, what does it mean? Are there previous studies in Indonesia or another LMICs that authors can talk about?

• Page 3: “Our previous observation showed that Indonesian adults were relatively short compared to WHO 2007 reference.12”. Is this mismatch purely genetic? Or is it the result of generational undernutrition?

Studies using archeological evidence like bones of ancient individuals living in India (11,000 years before the present and in the Mesolithic time),1,2 showed that these communities were taller than Europeans, and there has been a reduction in height through the centuries as a result of climatic adaptations, the transition to agriculture, and exposure to nutritional deficits specially during the British colonization (19th and 20th century), factors that might have contributed to the current growth faltering in India. I also understand from your previous findings on adult height in Indonesia3 that proxies of socioeconomic factors such educational attainment and share of household expenditure are associated positively with a higher height in adult life. Thus, this demonstrates that beyond genetic characteristics, there are macrolevel factors that might be associated with growth faltering in Indonesia.

Evidence from GWS (genome-wide association studies) have also shown that between 40-50% of height is genetically determined, especially among people from European ancestry;4,5 it seems that among people from other ancestries could be a lower percentage, but more information is needed. This suggests that there remaining percentage might be determined by nutrition and other environmental factors, that have key role at influencing growth and the final attained size since early life and across the life course.

Therefore, based on these findings and acknowledging that the WHO growth standards showed solid evidence on the potential to grow of all humans, independent of their genetic and ethnic background, if children have adequate conditions (food, support, are free of disease) to grow and develop, they will have a similar potential to attain the same size as they grow older, especially during the first 5 years of life.

In summary, I wonder what the rationale and evidence is to support the hypothesis that growth deceleration towards the genetic target height is the reason behind the downward shift in HAZ distribution instead of growth faltering across the entire population of children younger than 5 years of age. It is not clear from the evidence presented what would be the limitations of the WHO growth standards, and why the shifts in the distribution of HAZ among Indonesian children are not a reflection of growth faltering across the entire population (a common phenomenon in low-and middle-income countries, as Roth et al 2017 showed in an analysis of data from 64 LMICs).

References

1.Lukacs, J. R., & Pal, J. N. Skeletal Variation among Mesolithic People of the Ganga Plains: New Evidence of Habitual Activity and Adaptation to Climate. Asian Perspectives. 2003 42(2), 329–351.

2.Pomeroy E, Mushrif-Tripathy V, Cole TJ, Wells JCK, Stock JT. Ancient origins of low lean mass among South Asians and implications for modern type 2 diabetes susceptibility. Sci Rep. 2019 Jul 19;9(1):10515.

3.Moelyo AG, Sitaresmi MN, Julia M. Secular trends in Javanese adult height: the roles of environment and educational attainment. BMC Public Health. 2022 Apr 11;22(1):712.

4.Yengo, L., Vedantam, S., Marouli, E. et al. A saturated map of common genetic variants associated with human height. Nature 610, 704–712 (2022).

5.Yang J, Benyamin B, McEvoy BP, Gordon S, Henders AK, Nyholt DR, Madden PA, Heath AC, Martin NG, Montgomery GW, Goddard ME, Visscher PM. Common SNPs explain a large proportion of the heritability for human height. Nat Genet. 2010 Jul;42(7):565-9.

Methods

• The manuscript is missing some important details in the method sections based on STROBE guidelines for cross-sectional studies. Please provide more details about the data collection and for each variable of interest, give sources of data and details of methods of assessment (measurement). How did they get information about parents? Was the anthropometric data measured or self-reported in children and parents? Please provide information about the selection of the analytic sample.

• Statistical analyses can be improved. Authors only did a characterization of analytic sample and stratified by sociodemographic characteristics and showed differences in targeted height by these characteristics. I wonder why authors did not run regression analyses to test for differences among groups but at the same time to control for potential confounders (nutrition deficiencies, socioeconomic status, illness). Did the analyses include survey weights and other variables (cluster and stratum) that consider the survey design? (Important aspects to be able to make generalizations to the population level)

• I wonder what the proportion of stunting in your study population is? It would be interesting to do stratify analysis by children with normal length at birth vs children with stunting at birth, and/or to control for birth weight and birth length.

Discussion

• Not sure if authors can make conclusions that the differences in height-for-age z-scores between children and their parents (genetic targets) are a natural adaptation to genetic height. In this analysis, authors did not control for socioeconomic status, and other potential confounders. With the information provided, I am not sure if the study population had children with good health and nutritional status. I also think, if children had stunting at birth, there is a possibility that they might growth faster in the first two years as a reflection of catch-up growth and then reach their expected trajectory as they grow older.

• Page 7: “When the parents' height increases, so will the child's target height and HAZ.” Please add a reference to support this statement. In contrast to this statement, other authors have also shown that children with parents of short height will have the potential to attain a taller size. There is a secular trend of increments in height in the recent generations at the population level, especially in LMICs. (Cole TJ. The secular trend in human physical growth: a biological view. Econ Hum Biol. 2003 Jun;1(2):161-8)

• Page 8: Our study found an interesting finding in overweight–obese children. They decelerated

earlier to rebound earlier and higher compared to the other nutritional status group. Due to the

• cross sectional nature of our data, it was difficult to explain this phenomenon. This phenomenon has already been described in other populations using birth cohorts. Additionally, analyses on children with obesity seem disconnected from the study aim and research question. It should be tightly connected throughout the paper and included in the introduction.

• The potential limitations of this analysis are missing in the discussion section.

Minor comments

Introduction – Page 2: “The causes of the observed low mean length or height z-scores (HAZ) and the high prevalence of stunting in those countries have been subjected to many epidemiological studies, primarily focusing on individual or household-level risk factors.2”

Please be careful, this sentence seems like a replica of the original article (reference 3)

Reviewer #2: Moelyo and colleagues use large population data of children from several waves of the Indonesia Family Life Survey to examine linear growth declaration relative to parental target height. Research, generally of merit with lots of strengths. Some specific comments:

1. Introduction - generally clear although lacking current background data on the burden of the issue and related health consequences in your population. Would help if current prevalence of 'stunting'/growth faltering is provided as reference # 4 is based IFLS data upto 2007, rather outdated data to quantify population burden in present day Indonesia.

2. Methods. Lacking some description on anthropometric measurements and their quality, for example who measured height data? were they trained on standardized protocols? What was done to ensure reliable measurements over the 4 cycles of data used in this study? Such information could help address quality assurance issues.

3. Construct of target height. Not sure the use of THZ is clear as discerned here as data is serial cross-sectional data of different children. Parental height is their adult height, their own height during childhood is not available in this dataset so how are authors connecting child HAZ to parental THZ here? Kindly clarify.

4.Also genetic potential of growing children is not achieved until adulthood, so quantifying/characterizing SD score changes as deceleration toward genetic target height without longitudinal data challenging as positive or negative differences in HAZ-THZ are meaningful as they are SD scores at different ages. Could this be further clarified?

5.Discussion:

Generally clear but as discerned from text, struggling to capture the fundamental construct of extrapolating parental height to diagnose linear growth faltering of their offspring during childhood. Would help if this is carefully discussed because growth faltering or 'stunting' is a proxy of many other factors (maternal status, nutrition, intra-uterine growth, intergenerational transfer of environmental insults etc. etc. ) with the role of genetics being weaker in early life and picking up more during adolescence through early adulthood.

6. PLOS authors have the option to publish the peer review history of their article (what does this mean? ). If published, this will include your full peer review and any attached files.

**Do you want your identity to be public for this peer review?** For information about this choice, including consent withdrawal, please see our Privacy Policy .

Reviewer #1: No

Reviewer #2: No

---

## [Author Response · Author response to Decision Letter 0]

3 Nov 2023

Dear Reviewers,

Thank you very much for the suggestions and comments for our manuscript. We have sent a file of responses to reviewers. Hopefully, we have answered all of the suggestion and also added some explanations on the file. We also have made some major and minor changes to comply the suggestions.

Best regards,

Annang Giri Moelyo

---

## [Decision Letter · Decision Letter 1]

19 Dec 2023

PONE-D-23-23734R1Growth faltering or deceleration toward target height: linear growth interpretation using WHO Growth Standard 2006 for Indonesian childrenPLOS ONE

Dear Dr. Moelyo,

Thank you for submitting your manuscript to PLOS ONE. After careful consideration, we feel that it has merit but does not fully meet PLOS ONE’s publication criteria as it currently stands. Therefore, we invite you to submit a revised version of the manuscript that addresses the points raised during the review process.

 The reviewers have made very thoughtful comments and suggestions. I highly recommend considering their notes and incorporating into a second revision of your manuscript. 

We look forward to receiving your revised manuscript.

Kind regards,

Rebecca Evelyn Jones-Antwi, PhD

Academic Editor

PLOS ONE

Journal Requirements:

Additional Editor Comments:

Dear Author,

We thank you for your improvements to the paper. The reviewers have some more suggestions to help further improve the paper. Please consider their comments and notes thoughtfully.

best,

Dr. Jones-Antwi

Reviewers' comments:

Reviewer's Responses to Questions

**Comments to the Author**

1. If the authors have adequately addressed your comments raised in a previous round of review and you feel that this manuscript is now acceptable for publication, you may indicate that here to bypass the “Comments to the Author” section, enter your conflict of interest statement in the “Confidential to Editor” section, and submit your "Accept" recommendation.

Reviewer #1: All comments have been addressed

Reviewer #2: (No Response)

2. Is the manuscript technically sound, and do the data support the conclusions?

Reviewer #1: No

Reviewer #2: Yes

3. Has the statistical analysis been performed appropriately and rigorously? 

Reviewer #1: Yes

Reviewer #2: Yes

4. Have the authors made all data underlying the findings in their manuscript fully available?

Reviewer #1: Yes

Reviewer #2: Yes

5. Is the manuscript presented in an intelligible fashion and written in standard English?

Reviewer #1: Yes

Reviewer #2: Yes

6. Review Comments to the Author

Reviewer #1: Thank you to the authors for kindly reviewing and giving answers to the concerns raised by the reviewers. I think this topic is technically sound and the paper is trying to address an important scientific research question. However, the data and analyses presented do not support the conclusions that authors make: “We observed a deceleration toward target height, not growth faltering, in the first two years of the life of Indonesian children.” Of course, the deceleration towards the target height is observed, but it is not clear how this finding can rule out the possibility of growth faltering at the population level. As I mentioned in my previous review, the authors hypothesized that this negative shift of the distribution is not a reflection of growth faltering, but a deceleration towards their genetic target height (their parents attained height in adult life). It is scientifically sound to explore the causes and/or possible factors behind this population shift of the HAZ distribution among children under 5 in Indonesia, however, the evidence presented is not strong and clear enough to support that deceleration towards the genetic target height is the reason behind this distribution shift, and to rule out growth faltering across the whole population as the cause, as has been suggested in previous analysis.

Additionally, I disagree with authors who state that target height is not an individual level characteristic. This is an individual characteristic that they are evaluating at the population level through different surveys over the years. It is the genetic potential of children to attain certain size given their parents’ size; potential that is greater as children grow older (as observed in the graphs of older children and beta coefficient in supplementary tables). Regression analyses also showed that THz is one of the strongest predictors of HAZ, but household shares are also important predictors, and together with other predictors (Suppl Tables 2 a, b, and c) only explained 12-17% of the variability in HAZ (as shown by the adjusted R2). This means, that there might be other predictors that explain most of the variability in HAZ in children under five, beyond THZ and the characteristics included in these regressions.

Reviewer #2: The concept of target height has well known and clarified in the revised manuscript. I have a concern about using WHO 2007 at 19 years for estimating - THz given we know the WHO 2007 chart was based US-NHANES children (see ref # 22) and defeats the purpose potential adaptation to target height in their setting. Would be worth for authors to use internal z-scores (standardize your THz = (x-mean TH)/sd of TH) based the Indonesian parents in their database for each child, instead of WHO. Results will be in z-score units and offers a more objective way of examining the growth of these children in the environment their patients grew in.

7. PLOS authors have the option to publish the peer review history of their article (what does this mean? ). If published, this will include your full peer review and any attached files.

**Do you want your identity to be public for this peer review?** For information about this choice, including consent withdrawal, please see our Privacy Policy .

Reviewer #1: No

Reviewer #2: No

---

## [Author Response · Author response to Decision Letter 1]

30 Jan 2024

Responses to reviewer #1:

Thank you very much for the recommendation. We tried to revise the manuscript to make a better interpretation of what was observed. We just hope that we did not misunderstand your recommendations.

We agree that we cannot rule out the possibility of some incidences of growth faltering among the children. However, we argued that these incidences were more likely not the cause of the whole population distribution shift that could be seen consistently across the waves.

As suggested by the other reviewer, we calculated internal z-scores of the lengths/ heights of the children (age and sex adjusted) and of the target heights. The discrepancies between HAZ and THz (both calculated internally) are shown in the following figure.

The figure shows that the discrepancies between HAZ and THz (both calculated internally) are around zero. The mean discrepancy in waves 1993, 2000, 2007, and 2014 were (-0.18); (-0.07); 0.05; 0.09, respectively (p<0.05, One-way ANOVA test).

As the discrepancies in waves year 1993 and 2000 were negative (HAZ slightly lower than THz), there was a possibility that children at that time grew worse than did their parents when they were young. However, in later years, the discrepancies were positive, meaning children of waves 2007 and 2014 grew better than did their parents.

As for your second concern: The regression analyses showed that the models can only explain 12-17% of the variation in HAZ (although THz contributed the most).

This was the reason why we were refraining from interpreting our data at an individual level. Of course, an individual child’s length/ height would be influenced by thousands of variables that the available secondary data could not provide (See Bogin B, 2021. Social-Economic-Political-Emotional (SEPE) Factors Regulate Human Growth). What we can report in this paper is a consistent population level shift in the distribution of HAZ in relation to the distribution of target heights.

Reviewer #2:

Thank you very much for the recommendation. You recommend to use internal z-scores (standardized target height) based on the Indonesian parents themselves for each child, instead of WHO 2007.

When we calculated the internal z-scores of the target heights, we observed that the values of the z-scores were scattered around zero. Naturally, as might be expected, the mean of a normally distributed data would have a z-score of zero (Figure 2). As the mean z-scores of the children (Table 1), when compared to the international reference (i.e. WHO growth standard 2006), were between - 1.42 to -1.78, comparing the two z-sores seemed to be inappropriate.

Next, we calculated the internal z-scores of the children (sex and age adjusted), so the mean of their z-scores will also be zero. The discrepancy between HAZ and THz (both calculated internally), are shown in the following figure.

The figure shows that the discrepancies between HAZ and THz (both calculated internally) are around zero. The mean discrepancy in wave 1993, 2000, 2007, 2014 were (-0.18); (-0.07); 0.05; 0.09, respectively (p<0.05, One-way ANOVA test).

As the discrepancies in waves year 1993 and 1997 were negative (HAZ slightly lower than THz), there was possibility that children at that time grew worse than did their parents when they were young. However, at later years, the discrepancies were positive, meaning children of wave 2007 and 2014 grew better than did their parents. This is an interesting finding, however it is not the objective of our study

The objective of our study is to see how Indonesian children grow (as compared to international reference) when target heights were considered in the interpretation. Despite being based on US-NHANES children, the WHO 2007 chart was already designed to smoothly conform to WHO standard 2006. That is why we consider WHO 2007 chart as the continuation of the WHO 2006 growth standard, and use it to calculate z-scores of the target height.

---

## [Decision Letter · Decision Letter 2]

27 May 2024

PONE-D-23-23734R2Growth faltering or deceleration toward target height: 

linear growth interpretation using WHO Growth Standard 2006 for Indonesian childrenPLOS ONE

Dear Dr. Moelyo,

Thank you for submitting your manuscript to PLOS ONE. After careful consideration, we feel that it has merit but does not fully meet PLOS ONE’s publication criteria as it currently stands. Therefore, we invite you to submit a revised version of the manuscript that addresses the points raised during the review process.

We look forward to receiving your revised manuscript.

Kind regards,

Jay Saha

Academic Editor

PLOS ONE

Reviewers' comments:

Reviewer's Responses to Questions

**Comments to the Author**

1. If the authors have adequately addressed your comments raised in a previous round of review and you feel that this manuscript is now acceptable for publication, you may indicate that here to bypass the “Comments to the Author” section, enter your conflict of interest statement in the “Confidential to Editor” section, and submit your "Accept" recommendation.

Reviewer #2: All comments have been addressed

Reviewer #3: (No Response)

2. Is the manuscript technically sound, and do the data support the conclusions?

Reviewer #2: Yes

Reviewer #3: Partly

3. Has the statistical analysis been performed appropriately and rigorously? 

Reviewer #2: Yes

Reviewer #3: No

4. Have the authors made all data underlying the findings in their manuscript fully available?

Reviewer #2: Yes

Reviewer #3: No

5. Is the manuscript presented in an intelligible fashion and written in standard English?

Reviewer #2: Yes

Reviewer #3: Yes

6. Review Comments to the Author

Reviewer #2: No additional comments. Concerns satisfactorily addressed by authors. --------------------------------------------------------\\\\\\\\

Reviewer #3: 1. They state the height for age z-scores of the child tends to approach the target height irrespective of the birth z score being above or below what is predicted. The authors claim is that there is a difference between “faltering of growth” due to environmental factors and “deceleration toward target height” due to a physiological set point for height that is determined by the parents heights. And they further claim that short stature in Indonesia is due to short parents, and not necessarily conditions that have an adverse effect on child growth. Their main evidence of this is that children tend to be born at a length greater than expected, and across multiple socio-economic groups, although the latter is not actually shown in the data.

2. But Figure 1 shows that children are typically born at a z-score greater than predicted by the parents height, and they then approach the predicted z-score as they get older; but for the 4 cohorts from 1993 to 2014, the height of the children progressively exceeds the predicted height. These data suggest that intrauterine growth is better than predicted based on maternal and paternal height, and would indicate that the conditions in pregnancy have improved since the time the parents were children, as would be expected. And that postnatal conditions have also improved, but not as much, as evidenced by the children exceeding the target height in the more recent cohorts.

3. Figure 2 shows clearly that the more the children exceed the target height at birth, the taller they tend to stay. The fact that all the children decline in height indicates that for most children the post-natal growth conditions are suboptimal, not that there is a target height. While it is true that the degree of decline tends to be similar for the various trajectories that start from birth at higher than predicted z-scores, this does not mean there is not growth faltering, but rather indicates there is growth both growth faltering and an impact of parental height on child height, the latter of which is well known for 40 years or more.

4. To indicate there is a target height, it would also be useful to examine the trajectories of children born below the target height and to see if they have catch-up growth to the target height, especially after adjustment for socioeconomic status.

5. The authors should also show the trajectories of each child cohort based on the actual child z-scores and not only the deviations from the predicted z-scores. And they should show the trajectories stratified by socioeconomic status, by deciles.

6. They should also show the scatterplot and regression analysis, and variance explained, in child height based on parental heights.

7. Overall, the data are most parsimoniously explained by points #3 and #4, and not by birth length being programmed to descend to the target height determined by the parents heights, and that therefore growth faltering is absent. The authors should also be mindful that the stunting of any specific child is difficult to state based on a single measurement, and that growth trajectories of each child are the most useful. As such, stunting in a population is the important indicator, as a healthy population would be expected to have approximately 2.5% of the children below -2 SD (De Onis et al., 2019).

8. The authors should review the literature on growth faltering and the basis for the changes in height seen in multiple populations that experience economic development. Data from Indonesia show that children from high socioeconomic households are indeed reaching increased heights with progressive increases in z-scores, indicating that growth faltering in Indonesia is decreasing, as would be expected given its economic development. As such, “deceleration toward target height” which has been previously proposed by many persons is lacking in evidence when it has been examined in detail, and it remains a fallacy based on known trends in height of children and of populations, and especially for children from impoverished situations who were later raised under proper conditions wherein they routinely exceed growth expectations. While there is clearly a hereditary component of height, factors other than that account for a substantial proportion of stunting in populations with high prevalence stunted children.

9. The authors should do the analysis that has been suggested. The authors can utilize the IFLS data, that is a longitudinal survey with various social, economic, health, and child growth data.

7. PLOS authors have the option to publish the peer review history of their article (what does this mean? ). If published, this will include your full peer review and any attached files.

**Do you want your identity to be public for this peer review?** For information about this choice, including consent withdrawal, please see our Privacy Policy .

Reviewer #2: No

Reviewer #3: No

---

## [Author Response · Author response to Decision Letter 2]

30 Jul 2024

Responses to reviewer #3

1. They state the height for age z-scores of the child tends to approach the target height irrespective of the birth z score being above or below what is predicted.

The authors claim is that there is a difference between “faltering of growth” due to environmental factors and “deceleration toward target height” due to a physiological set point for height that is determined by the parents heights. And they further claim that short stature in Indonesia is due to short parents, and not necessarily conditions that have an adverse effect on child growth. Their main evidence of this is that children tend to be born at a length greater than expected, and across multiple socio-economic groups, although the latter is not actually shown in the data.

Response to #1:

Thank you very much for the recommendations. We will try to response to your inquiries and, if applicable, to revise the manuscript accordingly.

Previous study by Dwipoerwantoro PG et al (2015) (reference no. 9 in our manuscript), had criticized that the WHO growth standards did not reflect the growth of the present cohort of Indonesian infants and may overestimate the levels of underweight and stunted children. Based on this observation, we tried to find out whether target height was the factor we should consider. Previous study by Leroy et al (2014) also mentioned that height deficit occurred in all regions, with different size of the deficit.

Our study was mainly inspired by the paper of Roth et al., in 2017. In the paper, Roth et al showed that the downward shift of growth trajectory observed when the growth of children in the developing countries were compared to the WHO 2006 growth standard, affected both the upper and the lower end of the height distributions. As they grew, the taller children ‘lost’ height standard deviation scores as much as those of the shorter children.

Due to this parallel downward shift, Roth et al also suggested to, instead of looking at the individual or household level determinants of growth, to look for community level determinants that influence both the taller and the shorter children simultaneously.

To show our claim that this downward shift in height-for-age z-scores affected the entire spectrum of the HAZ distribution, i.e across multiple socioeconomic status groups, we will add the following figures as supplementary figures, completing the existing figures.

Supplementary Figure 2. Height-for-age z scores stratified by waves of the data collection (year 1993, 2000, 2007 and 2014). Similar patterns were observed across waves.

Supplementary Figure 3. Height-for-age z scores stratified by quartiles of the household expenditure shares. Quartile 1 were children from the poorest households while quartile 4 were children from the richest households. Similar patterns, albeit of different magnitudes, were observed across all socioeconomic status groups, at least until the age of two years.

Supplementary Figure 4a and 4 b. Height-for-age z scores stratified by paternal and maternal highest educational achievement. Similar downward patterns were observed between children of highly educated parents and children of lowly educated parents, at least until the age of 2 years.

Supplementary Figure 5. Height-for-age z scores stratified by urban or rural areas of residence. Similar downward patterns were also observed between children who lived in the rural area and those who lived in the urban.

Supplementary Figure 6. Height-for-age z scores stratified by whether the children lived in Java/ Bali islands or outer Java/Bali. Islands of Java and Bali were considered more developed than those of outer Java/ Bali. Similar downward patterns were also observed.

The existing figures (Fig. 1 and Fig 2, as well as supplementary Fig. 1) have also shown that the discrepancies between target heights and the children’s current HAZ were consistent across waves and urban/ rural residences.

Figure 2, interestingly, showed that the lower the target heights, the bigger was the positive discrepancy between HAZ and THz, meaning children of relatively shorter parents had the potential to be taller than their parents, but not children of relatively taller parents.

We can add figures on the discrepancies between THz and HAZ stratified by quartiles of the household expenditure shares, parental educational achievements, and living in Java/Bali or not if it is deemed necessary.

Furthermore, Supplementary table 2a, 2b, 2c showed that target height had the largest coefficient compared to the other predictors of height, i.e household expenditure share, parental education, lived in Java-Bali islands or outer, and lived in urban/rural areas.

2. But Figure 1 shows that children are typically born at a z-score greater than predicted by the parents height, and they then approach the predicted z-score as they get older; but for the 4 cohorts from 1993 to 2014, the height of the children progressively exceeds the predicted height. These data suggest that intrauterine growth is better than predicted based on maternal and paternal height, and would indicate that the conditions in pregnancy have improved since the time the parents were children, as would be expected. And that postnatal conditions have also improved, but not as much, as evidenced by the children exceeding the target height in the more recent cohorts.

Response to #2:

We agree with the reviewers’ comments. Indeed, the height of the children progressively exceeds the predicted height. This observation is in concordance with our previous paper on the secular trends in heights of adults in Indonesia (Moelyo et al., 2022, reference no. 25). Indonesian’s final height increased approximately 1 cm every generation (men and women: 1.3 cm and 0.9 cm per decade) (in approximately 20 years).

We have added this information into the discussion section (line 226) (highlighted).

3. Figure 2 shows clearly that the more the children exceed the target height at birth, the taller they tend to stay. The fact that all the children decline in height indicates that for most children the post-natal growth conditions are suboptimal, not that there is a target height. While it is true that the degree of decline tends to be similar for the various trajectories that start from birth at higher than predicted z-scores, this does not mean there is not growth faltering, but rather indicates there is growth both growth faltering and an impact of parental height on child height, the latter of which is well known for 40 years or more.

Response to #3:

We agree with the reviewer’s comments. Interestingly, Supplementary table 2a, 2b, 2c showed that target height had the largest coefficient compared to the other predictors of height, i.e household expenditure share, parental education, lived in Java-Bali islands or outer, and lived in urban/rural areas.

It seemed that parental heights were better predictors of the children’s current heights compared to the other predictors observed in this study. Parental heights should be considered before labelling growth faltering or stunting on an individual child.

4. To indicate there is a target height, it would also be useful to examine the trajectories of children born below the target height and to see if they have catch-up growth to the target height, especially after adjustment for socioeconomic status.

Response to #4:

Thank you for the recommendation. It became one of the limitations of our study. Our data is cross sectional repeated surveys with 7 years gap between the surveys. Hence, we are sorry that we cannot show the growth trajectory of any individual child. We have stated this as the limitations of our study in line 272-278.

5. The authors should also show the trajectories of each child cohort based on the actual child z-scores and not only the deviations from the predicted z-scores. And they should show the trajectories stratified by socioeconomic status, by deciles.

Response to #5:

We are sorry that we cannot show the growth trajectory of any individual child because of the nature of our data. Supplementary figures 2,3,4,5 and 6 showed that better socioeconomic status showed better heights, however the deceleration patterns of height z-scores were similar.

6. They should also show the scatterplot and regression analysis, and variance explained, in child height based on parental heights.

Response to #6:

We provided the scatterplot below (Must we put this also in supplementary figure?). We showed the multivariate analysis in supplementary tables 2 a, b and c and showed that target height z-scores had the largest coefficient regression compared to the others.

Fig. Scatterplot and Regression line between HAZ and THz

7. Overall, the data are most parsimoniously explained by points #3 and #4, and not by birth length being programmed to descend to the target height determined by the parents heights, and that therefore growth faltering is absent. The authors should also be mindful that the stunting of any specific child is difficult to state based on a single measurement, and that growth trajectories of each child are the most useful. As such, stunting in a population is the important indicator, as a healthy population would be expected to have approximately 2.5% of the children below -2 SD (De Onis et al., 2019).

Response to #7:

We completely agree that stunting of any specific child is difficult to state based on a single measurement, and that growth trajectory of each child are the most useful. However, this study suggested that considering parental heights when diagnosing stunting in an individual child, was very important.

We also completely agree that a healthy population is expected to have approximately 2.5% of the children below -2 SD. However, as has been questioned by many countries who do not use the WHO Growth Standard 2006 to assess their children’s growths, whether the use of only the WHO Growth Standard 2006 to judge the growth of Indonesian children is appropriate. This is because using only the WHO Growth Standard 2006 to judge the growth of an individual child will ignore the influence of parental heights.

In this study we only reported the heights of Indonesian children in relation to their parents’ heights, and we clearly showed that parental heights influenced the children heights across times and socioeconomic status.

8. The authors should review the literature on growth faltering and the basis for the changes in height seen in multiple populations that experience economic development. Data from Indonesia show that children from high socioeconomic households are indeed reaching increased heights with progressive increases in z-scores, indicating that growth faltering in Indonesia is decreasing, as would be expected given its economic development. As such, “deceleration toward target height” which has been previously proposed by many persons is lacking in evidence when it has been examined in detail, and it remains a fallacy based on known trends in height of children and of populations, and especially for children from impoverished situations who were later raised under proper conditions wherein they routinely exceed growth expectations. While there is clearly a hereditary component of height, factors other than that account for a substantial proportion of stunting in populations with high prevalence stunted children.

Response to #8:

Thank you for the explanation from the reviewer. Previous study by Roth et al (2017) (#reference 3) has reported the growth trajectories of children in low-income and middle-income countries. It concluded that the dominant underlying causes of postnatal linear growth faltering are population-wide exposures (ie, community-level or ubiquitous factors to which nearly all children in the population are exposed) rather than distinct behaviors or characteristics of certain individuals or households. This manuscript clearly showed that parental heights influenced the heights of the children, across times and socioeconomic status. Although, off course, other factors also had roles.

Our previous study showed the secular trend of Indonesian adult height. The birth year, educational attainment, and share of household food expenditure significantly influenced adult heights (Moelyo et al., 2022).

9. The authors should do the analysis that has been suggested. The authors can utilize the IFLS data, that is a longitudinal survey with various social, economic, health, and child growth data.

Response to #9:

Thank you for the recommendations. IFLS surveys were done every 7 years, so, although it is possible to use longitudinal information, the long gap made it impossible to assess only childhood growth. It would be impossible to find the trajectories in each year of age or each 6 months of age.

---

## [Decision Letter · Decision Letter 3]

21 Jan 2025

PONE-D-23-23734R3Growth faltering or deceleration toward target height: linear growth interpretation using WHO Growth Standard 2006 for Indonesian childrenPLOS ONE

Dear Dr. Moelyo,

Thank you for submitting your manuscript to PLOS ONE. After careful consideration, we feel that it has merit but does not fully meet PLOS ONE’s publication criteria as it currently stands. Therefore, we invite you to submit a revised version of the manuscript that addresses the points raised during the review process.

**ACADEMIC EDITOR: **
**It is a good and informative topic. Sorry for being late due to the unavailability of reviewers. Please go through the comments of the **
**reviewer and bring changes based on the comments.**

We look forward to receiving your revised manuscript.

Kind regards,

Bilal Ahmad Rahimi, M.D., D.T.M.&H., M.C.T.P., Ph.D

Academic Editor

PLOS ONE

Journal Requirements:

Reviewers' comments:

Reviewer's Responses to Questions

**Comments to the Author**

1. If the authors have adequately addressed your comments raised in a previous round of review and you feel that this manuscript is now acceptable for publication, you may indicate that here to bypass the “Comments to the Author” section, enter your conflict of interest statement in the “Confidential to Editor” section, and submit your "Accept" recommendation.

Reviewer #4: All comments have been addressed

2. Is the manuscript technically sound, and do the data support the conclusions?

Reviewer #4: Partly

3. Has the statistical analysis been performed appropriately and rigorously? 

Reviewer #4: Yes

4. Have the authors made all data underlying the findings in their manuscript fully available?

Reviewer #4: Yes

5. Is the manuscript presented in an intelligible fashion and written in standard English?

Reviewer #4: Yes

6. Review Comments to the Author

Reviewer #4: general view is of accepting after making these minor updates, for detail see the comments attached as document

7. PLOS authors have the option to publish the peer review history of their article (what does this mean? ). If published, this will include your full peer review and any attached files.

**Do you want your identity to be public for this peer review?** For information about this choice, including consent withdrawal, please see our Privacy Policy .

Reviewer #4: **Yes: ** Muhammad Aasim

---

## [Author Response · Author response to Decision Letter 3]

4 Mar 2025

Dear reviewer and editor,

We have made respond to reviewers and editor comments in response to reviewers file. We add explanations to ensure the manuscript technically sound, and do the data support the conclusions.

Best regards,

Annang Giri Moelyo

---

## [Editor Report · Decision Letter 4]

5 Mar 2025

Growth faltering or deceleration toward target height: linear growth interpretation using WHO Growth Standard 2006  for Indonesian children

PONE-D-23-23734R4

Dear Dr. Annang,

We’re pleased to inform you that your manuscript has been judged scientifically suitable for publication and will be formally accepted for publication once it meets all outstanding technical requirements.

Kind regards,

Bilal Ahmad Rahimi, M.D., D.T.M.&H., M.C.T.P., Ph.D

Academic Editor

PLOS ONE
---

## [Editor Report · Acceptance letter]

PONE-D-23-23734R4

PLOS ONE

Dear Dr. Moelyo,

I'm pleased to inform you that your manuscript has been deemed suitable for publication in PLOS ONE. Congratulations! Your manuscript is now being handed over to our production team.

Kind regards,

on behalf of

Professor Bilal Ahmad Rahimi

Academic Editor

PLOS ONE